# Diffusion Self-Weighted Guidance for Offline Reinforcement Learning

**Augusto Tagle**  *augustotaglemontes@gmail.com*
*Initiative for Data & AI, Universidad de Chile*

**Javier Ruiz-del-Solar**  *jruizd@ing.uchile.cl*
*AMTC & Dept. of Electrical Eng., Universidad de Chile*

**Felipe Tobar**  *f.tobar@imperial.ac.uk*
*Department of Mathematics, Imperial College London*

**Reviewed on OpenReview:** *https://openreview.net/forum?id=jmXBnpmznv*

## Abstract

Offline reinforcement learning (RL) recovers the optimal policy $\pi$ given historical observations of an agent. In practice, $\pi$ is modeled as a weighted version of the agent's behavior policy $\mu$, using a weight function $w$ working as a *critic* of the agent's behavior. Although recent approaches to offline RL based on diffusion models (DM) have exhibited promising results, they require training a separate guidance network to compute the required scores, which is challenging due to their dependence on the unknown $w$. In this work, we construct a diffusion model over both the actions and the weights, to explore a more streamlined DM-based approach to offline RL. With the proposed setting, the required scores are directly obtained from the diffusion model without learning additional networks. Our main conceptual contribution is a novel exact guidance method, where guidance comes from the same diffusion model; therefore, our proposal is termed *Self-Weighted Guidance* (SWG). Through an experimental proof of concept for SWG, we show that the proposed method i) generates samples from the desired distribution on toy examples, ii) performs competitively against state-of-the-art methods on D4RL when using resampling, and iii) exhibits robustness and scalability via ablation studies.

## 1 Introduction

Offline reinforcement learning (RL) learns a policy from a dataset of an agent's collected interactions. This setting is commonplace in scenarios such as robotics and healthcare, where real-world exploration with suboptimal agents is costly, dangerous, and thus unfeasible (Levine et al., 2020).

A critical issue when estimating the state-action value function $Q(s, \mathbf{a})$ (Q-values) in offline RL is the unknown effects of the actions absent in the dataset, referred to *out-of-sample actions*. In practice, methods such as (Kostrikov et al., 2022; Xu et al., 2023; Hansen-Estruch et al., 2023) refrain from querying out-of-sample actions during training, thus ensuring that the learned policy remains close to the dataset policy, and avoiding distributional shift (Kumar et al., 2019; Fujimoto et al., 2019; Nair et al., 2020). These methods typically adopt a *weighted behavior policy* form given by

$$\pi(\mathbf{a}|s) \propto \mu(\mathbf{a}|s)w(s, \mathbf{a}), \tag{1}$$

where $(s, \mathbf{a})$ is the state-action pair, $\pi(\mathbf{a}|s)$ is the target policy, $\mu(\mathbf{a}|s)$ is the dataset —or *behavior*— policy, and $w(s, \mathbf{a})$ is the weight function encoding the value of an action given a state.

Previous approaches learn $\pi(\mathbf{a}|s)$ in a weighted regression procedure (Peters & Schaal, 2007; Peng et al., 2019; Nair et al., 2020; Wang et al., 2020), using, e.g., a Gaussian model (Kostrikov et al., 2022); this

approach is severely limited by the expressiveness of the chosen model. For instance, when the dataset contains multimodal actions, unimodal policies will certainly fail to capture the agent's behavior successfully (Hansen-Estruch et al., 2023; Mao et al., 2024).

Diffusion models (DM) (Sohl-Dickstein et al., 2015; Ho et al., 2020; Song et al., 2021) have been successfully applied to sampling from the unnormalized target policy in equation 1 (Janner et al., 2022; Chen et al., 2023; Hansen-Estruch et al., 2023; Lu et al., 2023; Mao et al., 2024), while modeling a potentially multi-modal policy (Wang et al., 2022; Chi et al., 2025). This has been achieved by approximating the behavior policy using a DM, i.e., $\mu_\theta(\mathbf{a}|s) \approx \mu(\mathbf{a}|s)$, learning the weight function using external models (see Secs. 2.1 and 2.2), and then constructing a procedure to emulate samples from $\pi(\mathbf{a}|s)$. The main challenge of this approach is the estimation of the intractable score, where noisy actions and weights are entangled. This is typically addressed by learning an auxiliary model, referred to as the *guidance network* (Lu et al., 2023).

In this work, we propose a fundamentally different approach to diffusion-model-based offline RL. In addition to sampling actions from the behavior policy $\mu(\mathbf{a}|s)$, we also sample the weights $w(s, \mathbf{a})$ using the same DM. As shown in Sec. 3, this enables a sampling procedure that effectively generates actions from the target policy $\pi(\mathbf{a}|s)$. In contrast to previous approaches that rely on the training of two distinct models —a DM and an external guidance network— our method eliminates the need for an external guidance network by embedding the guidance information within the diffusion model itself. We refer to this central contribution as *Self-Weighted Guidance* (SWG), detailed in Sec.3.

The main objective of this work is to introduce SWG, a DM over actions and weights for offline RL, and validate its feasibility. Experimentally, we expect a competitive implementation of SWG which performs on par with the state-of-the-art. In particular, our contributions are summarized as follows.

- The derivation of the SWG method for exact guidance (Prop. 3.3 and Thm. 3.5 in Sec. 3).

- An implementation setup of weights models tailored for SWG (Sec. 5).

- A proof of concept for SWG's exact guidance capabilities in toy experiments (Sec. 6.1).

- An experimental validation of SWG in D4RL's challenging environments, where it exhibits competitive performance among recent methods and the ability to leverage a resampling stage (Sec. 6.2).

- Ablation studies exploring SWG's robustness and scaling properties (Secs. 6.3, 6.4 and 6.5).

## 2 Background

### 2.1 Offline reinforcement learning

Consider an agent that, following an unknown behavior policy $\mu(\mathbf{a}|s)$, interacted with the environment generating samples $(s, \mathbf{a}, r, s')$ recorded in a dataset $\mathcal{D}_\mu$. The goal of offline RL is to estimate the optimal policy $\pi(\mathbf{a}|s)$ from $\mathcal{D}_\mu$. Modeling the target policy as a weighted behavior policy in equation 1 is commonplace across various offline RL methods, some of which are presented next.

**Constrained policy optimization** (Peters et al., 2010; Peng et al., 2019; Nair et al., 2020) regularizes the learned policy to ensure that it remains close to the behavior policy by:

$$\pi^*(\mathbf{a}|s) = \arg\max_\pi \mathbb{E}_{s \sim D_\mu} \left[ \mathbb{E}_{\mathbf{a} \sim \pi(\mathbf{a}|s)} Q(s, \mathbf{a}) - \frac{1}{\beta} D_{KL}(\pi(\cdot|s) \| \mu(\cdot|s)) \right], \quad (2)$$

where $Q(s, \mathbf{a})$ is the state-action function that indicates the quality of an action given a state, in terms of the expected future discounted rewards under a specific policy.

Following (Peters et al., 2010; Peng et al., 2019; Nair et al., 2020), it can be shown that the optimal policy $\pi^*(\mathbf{a}|s)$ in equation 2 has a weighted structure as in equation 1, given by:

$$\pi^*(\mathbf{a}|s) \propto \mu(\mathbf{a}|s) e^{\beta Q(s, \mathbf{a})},$$

where $Q(s, \mathbf{a})$ represents the Q-values function of the optimal policy. Note that estimating the target policy Q-values requires querying out-of-sample actions.

**Implicit Q-learning (IQL)** (Kostrikov et al., 2022) avoids querying out-of-sample actions using an in-sample loss based on expectile regression. In this way, it defines an *implicit* policy that outperforms the behavior policy. Furthermore, IQL employs advantage-weighted regression (Peters & Schaal, 2007; Wang et al., 2018; Peng et al., 2019; Nair et al., 2020) to extract the policy, which can also be interpreted as a weighted behavior policy, given by:

$$\pi(\mathbf{a}|s) \propto \mu(\mathbf{a}|s)e^{\beta A(s,\mathbf{a})},$$

where $A(s, \mathbf{a}) = Q(s, \mathbf{a}) - V(s)$ is the advantage function learned with expectile regression.

**Implicit Diffusion Q-learning (IDQL)** (Hansen-Estruch et al., 2023) generalizes implicit formulations to the use of an arbitrarily convex function $f$:

$$V^*(s) = \arg\min_{V(s)} \mathbb{E}_{\mathbf{a}\sim\mu(\mathbf{a}|s)}\left[f(Q(s, \mathbf{a}) - V(s))\right],$$

thus defining a weighted behavior implicit policy given by

$$\pi_{\text{imp}}(\mathbf{a}|s) \propto \mu(\mathbf{a}|s)w(s, \mathbf{a}),$$

where $w(s, \mathbf{a}) = \frac{|f'(Q(s,\mathbf{a})-V^*(s))|}{|Q(s,\mathbf{a})-V^*(s)|}$. Furthermore, IDQL explores different convex functions $f$, such as expectiles (Kostrikov et al., 2022), quantiles (Koenker & Hallock, 2001) and exponential functions.

## 2.2 Diffusion models in offline RL

Diffusion models (DM) approximate an unknown data distribution[1] $q_0(\mathbf{a}_0)$ using a dataset $\mathcal{D}$ by defining a sequence of $k \in \{0, \dots, K\}, K \in \mathbb{N}$, noisy distributions defined as follows:

$$q_k(\mathbf{a}_k|\mathbf{a}_0) \sim \mathcal{N}(\mathbf{a}_k; \alpha_k\mathbf{a}_0, \sigma_k^2\mathbf{I}),$$

where $\alpha_k$ and $\sigma_k$ are predefined *noise schedules*, s.t., $\mathbf{a}_K \sim \mathcal{N}(0, \mathbf{I})$ as $K \to \infty$. Therefore, samples from $q_0$ can be obtained by first sampling from $\mathcal{N}(0, \mathbf{I})$, and then solving the reverse diffusion process. In practice, the reverse diffusion is approximated by a model $\epsilon_\theta$ that predicts the noise to be removed from a noisy sample $\mathbf{a}_K \sim q_K$, to produce a target sample $\mathbf{a}_0 \sim q_0$. The training loss for $\epsilon_\theta$ is (Ho et al., 2020)

$$\arg\min_\theta \mathbb{E}_{\mathbf{a}_0,\epsilon_0,k}||\epsilon_0 - \epsilon_\theta(\mathbf{a}_k, k)||_2^2, \tag{3}$$

where $\mathbf{a}_0 \sim q_0$, $\epsilon_0 \sim \mathcal{N}(0, \mathbf{I})$, $k \sim \mathcal{U}(1, K)$ and $\mathbf{a}_k = \alpha_k\mathbf{a}_0 + \sigma_k\epsilon_0$. The noise is related to the *score function* (Song et al., 2021) of the intermediate noisy distribution as follows:

$$\frac{-\epsilon_0}{\sigma_k} = \underbrace{\nabla_{\mathbf{a}_k}\log q_k(\mathbf{a}_k)}_{\text{Score function}}.$$

Our motivation for employing diffusion models (DMs) in offline RL stems from two main factors: (1) DMs are highly expressive compared to classical generative models (e.g., Mixture Models), allowing them to model multi-modal action distributions (Chi et al., 2025); and (2) their score-based formulation enables flexible sampling from conditioned or weighted distributions via guidance. In particular, the weighted behavior in equation 1 can be naturally incorporated using DMs, making them well suited for offline RL.

To condition the DM on the states, the noise prediction network can be made an explicit function of the state, that is, $\epsilon(\mathbf{a}_k, k) = \epsilon(\mathbf{a}_k, k, s)$. Previous approaches fit a DM $\mu_\theta(\mathbf{a}|s)$ to the behavior policy $\mu(\mathbf{a}|s)$, learn the weights $w(s, \mathbf{a})$ as described in Sec. 2.1, and then sample from the target policy via resampling or guidance; these are explained as follows.

---

[1]We denote data with the symbol $\mathbf{a}$ (instead of the standard $x$) for consistency, since in our setting the data are the agent's *actions*.

**Resampling** involves first sampling a batch of actions from the learned behavior policy $\mu_\theta(\mathbf{a}|s)$, and then resampling them using weights as probabilities of a categorical distribution (Hansen-Estruch et al., 2023; Chen et al., 2023), in a way inspired by importance sampling. Alternatively, one can adopt a greedy policy that simply selects the action that maximizes the weight with the aim of improving performance.

**Guidance** aims to obtain samples from the target policy by modifying the diffusion, adding a score term approximated by an external guidance network $g_\phi$ (Janner et al., 2022; Lu et al., 2023; Mao et al., 2024). The $k$-step score in a diffusion sampling procedure is given by:

$$\underbrace{\nabla_{\mathbf{a}_k} \log \pi_k(\mathbf{a}_k|s)}_{\text{target score}} \approx \underbrace{\nabla_{\mathbf{a}_k} \log \mu_\theta(\mathbf{a}_k, k, s)}_{\approx \frac{-\epsilon_\theta(\mathbf{a}_k, k, s)}{\sigma_k}} + \underbrace{\nabla_{\mathbf{a}_k} g_\phi(\mathbf{a}_k, k, s)}_{\text{guidance score}}, \tag{4}$$

where $\{\mathbf{a}_k\}_0^K$ are the noisy versions of the actions, with $\mathbf{a}_0$ being the true noise-free action. As shown by (Lu et al., 2023), an unbiased estimate of the guidance score that effectively produces samples from the target distribution should satisfy the following:

$$\nabla_{\mathbf{a}_k} g_\phi(\mathbf{a}_k, k, s) = \nabla_{\mathbf{a}_k} \log \mathbb{E}_{\mu(\mathbf{a}_0|\mathbf{a}_k, s)}[w(\mathbf{a}_0, s)], \tag{5}$$

where the right-hand term is intractable for most policies and weight functions. Guidance methods that are unbiased towards this objective are referred to as *exact guidance* methods.

Diffuser (Janner et al., 2022), trains the guidance network with a mean-squared-error objective with respect to Q-values, which produces a biased target policy, in the sense that it does not match the form in equation 5. Conversely, exact guidance methods such as Contrastive Energy Prediction (CEP) (Lu et al., 2023) use a contrastive learning objective that requires querying a batch of out-of-sample actions for each state in the dataset. Lastly, D-DICE (Mao et al., 2024), proposes an in-sample loss to learn the guidance network and uses a *guide-then-select* approach to obtain samples.

*Remark* 2.1. While exact guidance methods effectively produce samples from the target distribution, they require learning an external network $g_\phi$ through an intricate loss function $\mathcal{L}_g(\phi)$. This motivates our proposal *Self-Weighted Guidance*, which leverages DM's without resorting to a special training objective to conduct exact guidance.

## 3 Self-weighted guidance

### 3.1 Problem statement

Let us consider a dataset $\mathcal{D}_A = \{\mathbf{a}^{(i)}\}_{i=1}^N \subset \mathbb{R}^d$ of i.i.d. realizations of a random variable $A \sim q(\mathbf{a})$; these will be the agent's *actions*. Recall that we aim to sample from the unnormalized target $p(\cdot)$ that relates to $q(\cdot)$ as

$$p(\mathbf{a}) \propto q(\mathbf{a})w(\mathbf{a}), \tag{6}$$

where $w : \mathbb{R}^d \to \mathbb{R}$ is a known[2] weight function. Sampling can be achieved by fitting a DM, denoted $\epsilon_\theta$, to $q(\mathbf{a})$ trained on $\mathcal{D}_A$. Then, in the $k$-th denoising step, the score of the target distribution (referred to as the *target score*) satisfies:

$$\underbrace{\nabla_{\mathbf{a}_k} \log p_k(\mathbf{a}_k)}_{\text{target score}} = \underbrace{\nabla_{\mathbf{a}_k} \log q_k(\mathbf{a}_k)}_{\approx \frac{-\epsilon_\theta(\mathbf{a}_k, k)}{\sigma_k}} + \underbrace{\nabla_{\mathbf{a}_k} \log \mathbb{E}_{q(\mathbf{a}_0|\mathbf{a}_k)}[w(\mathbf{a}_0)]}_{\text{intractable score}}, \tag{7}$$

where $\mathbf{a}_0 \in \mathcal{D}_A$ is a data sample, $\mathbf{a}_1, \ldots, \mathbf{a}_K$ are noisy copies of the data on which the DM is trained, and the subindex $\cdot_k$ in the distributions denotes the denoising step. Although $\nabla_{\mathbf{a}_k} \log q_k(\mathbf{a}_k)$, referred to as *the intermediate score*, is provided directly by the DM, estimating the target score in equation 7 is challenging due to the intractability of the right-hand term in the general case (Lu et al., 2023).

---

[2]Though in offline RL the weight function $w$ has to be learned, we consider $w$ known for now.

## 3.2 Proposal: diffusing over actions and weights

Let us consider a random variable (RV) $Z = (A, W)$ given by the concatenation of $A$, describing the actions in the previous section, and $W$, which is given by passing $A$ through the weight function $w(\cdot)$. Since $w(\cdot)$ is a deterministic function of $A$, the distribution of $Z$ is given by the push-forward measure of the distribution of $A$ through the mapping $T : \mathbf{a} \mapsto (\mathbf{a}, w(\mathbf{a}))$. With a slight abuse of notation, we denote the push-forward measures corresponding to $p$ and $q$ in equation 6 with the same symbols, i.e. $p(\mathbf{z}) = T_\# p(\mathbf{a})$ and $q(\mathbf{z}) = T_\# q(\mathbf{a})$, respectively. Therefore, we have

$$p(\mathbf{z}) = p(\mathbf{a}, w) = p(w|\mathbf{a})p(\mathbf{a}) = p(\mathbf{a})\delta_{w=w(\mathbf{a})}$$
$$q(\mathbf{z}) = q(\mathbf{a}, w) = q(w|\mathbf{a})q(\mathbf{a}) = q(\mathbf{a})\delta_{w=w(\mathbf{a})},$$

which, combined with equation 6, give $p(\mathbf{z}) \propto q(\mathbf{z})w$. In this new formulation, $w$ is no longer to be considered as a function of $\mathbf{a}$, but rather as a variable for which there are available observations. For clarity of presentation, we define the following function

**Definition 3.1** (Extraction function). The function $\phi_w : \mathbb{R}^{d+1} \to \mathbb{R}$ extracts the weight component from the concatenated variable $\mathbf{z} = [\mathbf{a}, w]$. That is, $\phi_w : [\mathbf{a}, w] \mapsto \phi_w([\mathbf{a}, w]) = w$.

*Remark* 3.2. The extraction function allows us to express

$$p(\mathbf{z}) \propto q(\mathbf{z})\phi_w(\mathbf{z}), \tag{8}$$

thus implying that the target distribution for $\mathbf{z}$ follows the same structure as the original distribution in equation 6. This motivates jointly learning weights and actions with the same DM. Since the extraction function $\phi_w(\mathbf{z})$ is linear, it is feasible to estimate the (otherwise intractable) score within our DM.

We can now implement a DM to sample from the augmented variable $Z$. To this end, we first extend the dataset $\mathcal{D}_A = \{(\mathbf{a}^{(i)})\}_{i=1}^N$ to $\mathcal{D}_Z = \{(\mathbf{a}^{(i)}, w^{(i)})\}_{i=1}^N \subset \mathbb{R}^{d+1}$, where $w^{(i)} = w(\mathbf{a}^{(i)})$ is a weight observation. Notice that the dataset $\mathcal{D}_Z$ provides access to the weight function only on the support of the original data in $\mathcal{D}_A$. For consistency, we also denote the realizations of $Z$ as $\mathbf{z}^{(i)} = (\mathbf{a}^{(i)}, w^{(i)})$.

Therefore, we define a $K$-step diffusion with noise schedule $\{\alpha_k\}_{k=1}^K$, $\{\sigma_k\}_{k=1}^K$, and train a noise-prediction DM $\epsilon_\theta$ for $Z$ using the augmented dataset $\mathcal{D}_Z$, minimizing the standard diffusion loss in equation 3. In our notation, this is

$$\mathcal{L}(\theta) = \mathbb{E}_{\mathbf{z}_0, \epsilon_0, k} ||\epsilon_0 - \epsilon_\theta(\mathbf{z}_k, k)||_2^2, \tag{9}$$

where $\mathbf{z}_0 \sim \mathcal{D}_Z$, $\mathbf{z}_k = \alpha_k \mathbf{z}_0 + \sigma_k \epsilon_0$, $\epsilon_0 \sim \mathcal{N}(0, \mathbf{I})$, and $k \sim \mathcal{U}(1, K)$. Algorithm 1 describes our proposed DM training procedure. We emphasize that, in practice, Algorithm 1 is implemented with the expectile weight function.

---

**Algorithm 1** Training of joint DM over weights and actions

---

**Input:** Dataset $\mathcal{D}_A$, weight function $w(\cdot)$, number of diffusion steps $K$, noise schedule $\{\alpha_k\}_{k=1}^K$, $\{\sigma_k\}_{k=1}^K$, learning rate $\lambda$, and batch size $B$.
**Initialize:** $\theta$ parameters for the diffusion model $\epsilon_\theta$
Augment dataset $\mathcal{D}_A \to \mathcal{D}_Z$ as described in Sec. 3.2.
**for** each parameters update **do**
    Draw $B$ samples $\mathbf{z}_0 \sim \mathcal{D}_Z$ and $B$ indices $k \sim \mathcal{U}(1, K)$
    Add noise to each sampled data $\mathbf{z}_0$ for respective $k$ time as $\mathbf{z}_k = \alpha_k \mathbf{z}_0 + \sigma_k \epsilon_0$, $\epsilon_0 \sim \mathcal{N}(0, \mathbf{I})$
    Calculate loss $L(\theta) \approx \frac{1}{B} \sum_B ||\epsilon_0 - \epsilon_\theta(\mathbf{z}_k, k)||_2^2$
    Update parameters $\theta \leftarrow \theta - \lambda \nabla_\theta L(\theta)$
**end for**
**Return:** Trained diffusion model $\epsilon_\theta$

---

### 3.3 Augmented exact guidance

**Proposition 3.3.** *The required score to sample from the target distribution in equation 8 is given by*

$$\nabla_{\mathbf{z}_k} \log p_k(\mathbf{z}_k) = \underbrace{\nabla_{\mathbf{z}_k} \log q_k(\mathbf{z}_k)}_{\approx \frac{-\epsilon_\theta(\mathbf{z}_k, k)}{\sigma_k}} + \underbrace{\nabla_{\mathbf{z}_k} \log \mathbb{E}_{q(\mathbf{z}_0 | \mathbf{z}_k)} [\phi_w(\mathbf{z}_0)]}_{intractable\ term}. \tag{10}$$

The proof for Proposition 3.3 follows as an extension of (Lu et al., 2023) to our setting (see Appendix A.1).

*Remark* 3.4. Since the extended formulation in equation 8 is defined over $\mathbf{z}$, the gradients required to implement guidance in equation 10 are taken with respect to $\mathbf{z}_k$, rather than $\mathbf{a}_k$ (as in equation 7). This is a desired feature, since the actions and weights are highly coupled, and thus guiding them separately can lead to inefficient sampling.

The key insight of our proposal is that the intractable term in equation 10 can be recovered directly from the DM described above using the so-called *data prediction formula* (Efron, 2011; Kingma et al., 2021) given by:

$$\mathbb{E}_{q(\mathbf{z}_0 | \mathbf{z}_k)}[\mathbf{z}_0] \approx \mathbf{z}_\theta(\mathbf{z}_k, k) := \frac{\mathbf{z}_k - \sigma_k \epsilon_\theta(\mathbf{z}_k, k)}{\alpha_k}. \tag{11}$$

Using equation 11, and since the extraction function $\phi_w(\cdot)$ is linear, the target score $\nabla_{\mathbf{z}_k} \log p_k(\mathbf{z}_k)$ in equation 10 can be computed with the proposed DM, as stated in Theorem 3.5 (proof in Appendix A.2).

**Theorem 3.5.** *Let $\epsilon_\theta^*(\mathbf{z}_k, k)$ be the DM found as the global minimizer of equation 9. The exact target score of equation 10 is given by:*

$$\nabla_{\mathbf{z}_k} \log p_k(\mathbf{z}_k) = \underbrace{\nabla_{\mathbf{z}_k} \log q_k(\mathbf{z}_k)}_{\frac{-\epsilon_\theta^*(\mathbf{z}_k, k)}{\sigma_k}} + \underbrace{\nabla_{\mathbf{z}_k} \log \left( \phi_w \left( \frac{\mathbf{z}_k - \sigma_k \epsilon_\theta^*(\mathbf{z}_k, k)}{\alpha_k} \right) \right)}_{self\ guidance},$$

*where recall that $\phi_w(\mathbf{z}) = w$ simply extracts the weight component from $\mathbf{z} = [\mathbf{a}, w]$ (Def. 3.1).*

Our method is referred to as *Self-Weighted Guidance* (SWG), since the DM samples from the target distribution $\mathbf{z}_0 \sim p(\mathbf{z}_0)$ by *guiding itself*, that is, the guidance signal is a function of the same model (see Algorithm 2). SWG provides an elegant way of performing exact guidance, since it computes gradients directly on the DM $\epsilon_\theta$ instead of an external guidance network with a dedicated —and possibly intricate— loss function (see Sec. 4). This translates to not having to train a separate guidance network, thereby simplifying the training pipeline. Sec. 4 provides a theoretical comparison between SWG and other DM-based guidance approaches to offline RL, the results of which are summarized in Table 1.

---

**Algorithm 2** Self-Weighted Guidance sampling

**Input:** Pretrained DM $\epsilon_\theta$, number of diffusion steps $K$, and noise schedule $\{\alpha_k\}_{k=1}^K$, $\{\sigma_k\}_{k=1}^K$.
Sample noise $\mathbf{z}_k \sim \mathcal{N}(0, \mathbf{I})$
**for each** $k$ **in reversed** $0 : K$ **do**
    Forward pass on diffusion model $\epsilon_\theta(\mathbf{z}_k, k)$
    $\hat{\epsilon} := \epsilon_\theta(\mathbf{z}_k, k) - \sigma_k \nabla_{\mathbf{z}_k} \log \left( \phi_w \left( \frac{\mathbf{z}_k - \sigma_k \epsilon_\theta(\mathbf{z}_k, k)}{\alpha_k} \right) \right)$
    Calculate $\mathbf{z}_{k-1}$ with noise prediction $\hat{\epsilon}$
    Set $\mathbf{z}_k := \mathbf{z}_{k-1}$
**end for**
**Return:** Sample $\mathbf{z}_0$

---

## 4 Relationship to previous guidance methods in offline RL

In this section, we provide an analysis of previous guidance methods alongside our own, highlighting theoretical differences and potential advantages. The two primary criteria for comparing previous methods with SWG

are: i) whether the method provides exact guidance, and ii) whether it relies on an external guidance network, as summarized in Table 1.

Table 1: Summary of DM-based guidance approaches to offline RL.

| Method | Exact guidance | Requires External Guidance Network | Loss function |
|---|---|---|---|
| Diffuser (Janner et al., 2022) | No | Yes | Guidance network MSE over Q-values |
| CEP (Lu et al., 2023) | Yes | Yes | Contrastive loss (see equation 12) |
| D-DICE (Mao et al., 2024) | Yes | Yes | In-sample loss (see equation 13) |
| SWG (ours) | Yes | No | Diffusion loss (see equation 14) |

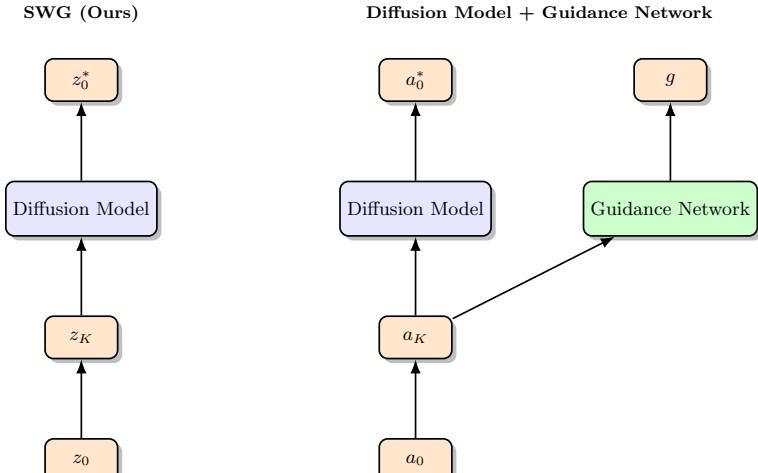

Figure 1: Comparison between the proposed SWG approach (left) and the standard Diffusion Model with a separate Guidance Network (right). SWG jointly learns actions and weights within a single diffusion model, whereas standard approaches in diffusion for RL trains an additional guidance network to steer the diffusion model. The figure represents one forward pass to the diffusion model during training. Recall that $\mathbf{z_0} = [\mathbf{a_0}, w_0]$.

Let us consider the two existing exact-guidance DM methods in the offline RL literature: CEP (Lu et al., 2023) and D-DICE (Mao et al., 2024). Assuming a known weight function $w(\cdot)$, CEP learns the guidance network $g_\theta$ using the loss

$$\mathcal{L}_{CEP}(\theta) = \mathbb{E}_{k, \mathbf{a}_{0,k}^{(1:M)}} \left[ -\sum_{i=1}^{M} w(s, \mathbf{a}_0^{(i)}) \log \frac{e^{g_{\theta,k}^{(i)}}}{\sum_{j=1}^{M} e^{g_{\theta,k}^{(j)}}} \right], \tag{12}$$

where $k \sim \mathcal{U}(1, K)$, and for simplicity of notation we denote $g_{\theta,k}^{(i)} = g_\theta(s, \mathbf{a}_k^{(i)}, k)$ and $\mathbf{a}_{0,k}^{(1:M)} = \{\mathbf{a}_j^{(i)}, \text{s.t.}, j \in \{0, k\}, i \in \{1, \ldots, M\}\}$. Recall that $\mathbf{a}_0$ is the data and $\mathbf{a}_k$ are the noisy copies. Since equation 12 requires $M > 1$ action samples from each state in the dataset, its evaluation is likely to require out-of-sample actions, thus possibly leading to overestimation errors (Kostrikov et al., 2022; Mao et al., 2024). CEP (Lu et al., 2023) addresses this by pretraining a DM on the behavior policy and generating a *synthetic* action dataset containing $M > 1$ samples for each state in the dataset.

In contrast, D-DICE proposes a loss that circumvents the need to query out-of-sample actions through

$$\mathcal{L}_{D-DICE}(\theta) = \mathbb{E}_{k, \mathbf{a}_{0,k}} \left[ w(s, \mathbf{a}_0) e^{-g_\theta(s, \mathbf{a}_k, k)} + g_\theta(s, \mathbf{a}_k, k) \right], \tag{13}$$

where $k \sim \mathcal{U}(1, K)$, $\mathbf{a}_0 \sim D$ is an action sample for the state $s$, and $\mathbf{a}_k$ is the respective noisy version of $\mathbf{a}_0$.

Although the objectives in equation 12 and equation 13 yield unbiased estimates of the intractable term in equation 5, their formulations introduce potential training instabilities arising from the exponential factor $e^{g_\theta}$. To address this, the authors of these loss functions proposed stabilization techniques: CEP employs *self-normalization* of the exponential term, whereas D-DICE alleviates the issue by clipping the exponential term. Furthermore, we contend that these loss functions are either lacking in direct interpretability, as in the case of D-DICE, or overly intricate, as in the case of CEP.

The proposed SWG method, conversely, is trained with the standard diffusion loss

$$\mathcal{L}_{SWG} = \mathbb{E}_{k, \mathbf{z}_0, \mathbf{z}_k} ||\epsilon_0 - \epsilon_\theta(\mathbf{z}_k, k)||_2^2, \tag{14}$$

which is equivalent to maximizing a variational lower bound on the likelihood (Kingma et al., 2021). That is, a simple MSE over predicted noise terms free from exponential terms prone to instabilities, thereby easing the method's implementation. Another relevant feature of SWG is that it inherently defines an in-sample loss, thus addressing the out-of-sample problem with the CEP objective.

Also, since SWG produces action and weight samples $\mathbf{z}_0 = [\mathbf{a}_0, w_0]$ simultaneously, it is possible to define a resampling mechanism as described in Sec 2.2 to further improve performance. In Section 6, where our experimental results are presented, we include a variant of SWG called SWG-R, which uses resampling. These results demonstrate that SWG-R exhibits superior performance in comparison to the vanilla (non-resampling) SWG method.

## 5 Implementing SWG in offline RL

There are two main practical aspects regarding the implementation of SWG in the offline RL setup. First, recall that, for notational and conceptual simplicity, the SWG was presented in a way independent of the states. In practice, however, state-dependent DM's are trained by considering the state as an input to the noise predictor network, that is, $\epsilon(\mathbf{z}_k, k) = \epsilon(\mathbf{z}_k, k, s)$ in equation 9. Second, SWG was introduced in Sec. 3 assuming a known weight function $w$, however, in offline RL the weights need to be estimated beforehand. We next present how to adapt and implement different weight formulations within the proposed SWG.

**Expectile.** The IDQL (Hansen-Estruch et al., 2023) $\tau$-expectile weight is defined as $w_\tau^2(s, \mathbf{a}) = \left| \tau - 1_{\{Q(s,\mathbf{a}) < V_\tau^2(s)\}} \right|$, where $V_\tau^2(s)$ is the value function learned with the expectile loss, and $1_{\{\cdot\}}$ is the indicator function. Since this weight takes only two possible values, $w_\tau^2(s, \mathbf{a}) \in \{\tau, 1 - \tau\} \subset [0, 1]$, it may fail to distinguish between similar actions by assigning them identical weights. To address this limitation, we propose a smooth variant of these weights, given by

$$w(s, \mathbf{a}) = \left( \frac{1}{1 + e^{-A(s,\mathbf{a})}} \right) \tau + \left( 1 - \frac{1}{1 + e^{-A(s,\mathbf{a})}} \right) (1 - \tau). \tag{15}$$

Here, we apply a sigmoid function to the advantage $A(s, \mathbf{a})$ to obtain a continuous weight function. This weight function is used in our D4RL experiments (Sec. 6.2), and the full training procedure for the value functions is detailed in Algorithm 3 in Appendix B.2.

**Exponential.** From the same value functions learned with expectile loss, we can define an exponential weight model derived from the IQL (Kostrikov et al., 2022) policy extraction method as $w(s, \mathbf{a}) = e^{\beta A(s,\mathbf{a})}$, where $A(s, \mathbf{a}) = Q(s, \mathbf{a}) - V(s)$ is the advantage function. Though this formulation addresses the sparsity issue arising in the expectile weight, its exponential form could lead to arbitrarily large values, and is thus potentially unstable during training of SWG. Using the advantage function can help mitigate instabilities, as the value function serves as a normalizer for the Q-values.

**Linex.** Another exponential formulation can be obtained when learning the value function using the linex function $lx : u \mapsto \exp(u) - u$. As found by (Hansen-Estruch et al., 2023), this defines the following weight:

$$w_{\exp}(s, \mathbf{a}) = \frac{\alpha \left| \exp\left( \alpha \left( Q(s, \mathbf{a}) - V_{\exp}(s) \right) \right) - 1 \right|}{\left| Q(s, \mathbf{a}) - V_{\exp}(s) \right|}, \tag{16}$$

where $V_{\exp}(s)$ is the value function learned by using the linex loss, and $\alpha$ is a temperature coefficient.

We emphasize that, among the various weighted behavior policies in the literature, we focus on avoiding querying out-of-sample actions, as they can lead to overestimation errors (Kostrikov et al., 2022). Sec. 6.3 provides an experimental comparison of different weight formulations within SWG on D4RL. More details on the implementation and training on the learning of $Q(s, a)$ and $V(s)$ are provided in Appendix B.2.

*Remark* 5.1. The standard practice across most offline RL methods is to estimate $Q(s, \mathbf{a})$ and $V(s)$ to then construct the weight function; this adds complexity to the overall training pipeline. We clarify that though SWG does not require an external guidance network during sampling, it still requires learning of the weight function prior to sampling.

In our implementation, we employ two techniques to refine the action generation process. First, we control the strength of the guidance score using a hyperparameter $\rho$, termed *guidance scale* (Dhariwal & Nichol, 2021), as follows $\hat{\epsilon} := \epsilon_\theta(\mathbf{z}_k, k) - \rho \sigma_k \nabla_{\mathbf{z}_k} \log \left(\phi_w\left(\frac{\mathbf{z}_k - \sigma_k \epsilon_\theta(\mathbf{z}_k, k)}{\alpha_k}\right)\right)$. Second, we introduce SWG-R, a resampling variant of SWG, which resamples a candidate batch of $M$ actions, denoted as $\{a_m\}_{m=0}^M$, and selects the action $a^*$ associated with the maximum weight $w$.

# 6 Experiments

The proposed SWG was implemented and validated through a set of experiments targeting the following questions:

1. Can SWG sample from the target distribution? (Sec. 6.1)

2. How does SWG (with and without resampling) perform against the state-of-the-art in the D4RL benchmark? (Sec. 6.2)

3. How do different weight formulations affect SWG's performance? (Sec. 6.3)

4. How does the guidance scale affect SWG's performance? (Sec. 6.4)

5. How does SWG's inference time scale with increasing model size? (Sec. 6.5)

Additional experimental details and benchmarks are given in Appendix B, with the full computational resources used in Appendix B.4. Our code is available at **SWG repository**.

## 6.1 Sampling from the target distribution

We considered the toy experiments from (Lu et al., 2023), which use a weight function $w(\mathbf{a}) = \exp(-\beta \xi(\mathbf{a}))$, where $\xi(\mathbf{a})$ is an energy function, and $\beta$ is the inverse temperature coefficient. Figures 2 and 3 show SWG samples from a spiral and an 8-Gaussian-mixture distributions, respectively, and offer a qualitative assessment of its performance. In both cases, samples closely matching the ground truth for different values of $\beta$ suggest that SWG was able to sample from the target distributions, thus learning the relationship between the data distribution and the weights in both cases. The reader is referred to the Appendix B.1 for further details and more examples with different target distributions.

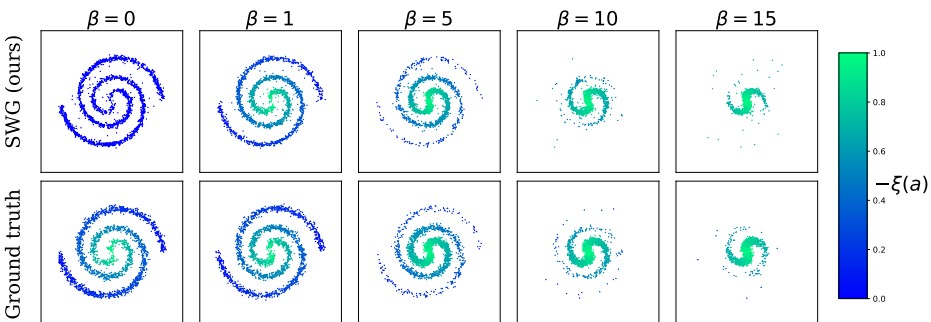

Figure 2: SWG applied to a spiral distribution for different temperature coefficients $\beta$. Top: samples generated by SWG. Bottom: ground truth samples. When $\beta = 0$, we have $w(\mathbf{a}) = 1$ and thus the target distribution matches the data distribution.

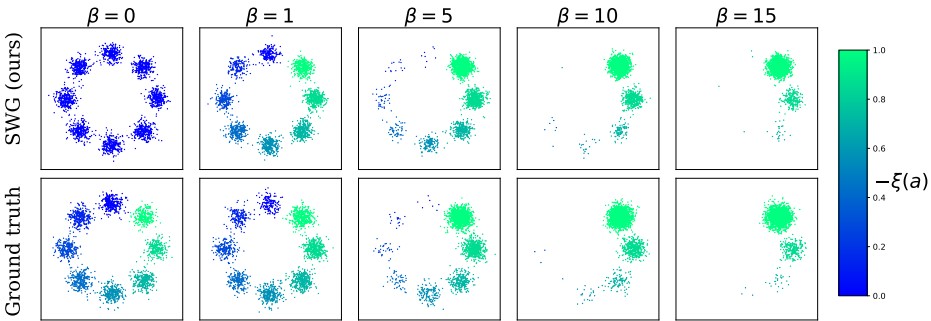

Figure 3: SWG applied to an 8-Gaussian mixture distribution for different temperature coefficients $\beta$. Top: samples generated by SWG. Bottom: ground truth samples. When $\beta = 0$, we have $w(\mathbf{a}) = 1$ and thus the target distribution matches the data distribution.

## 6.2 D4RL experiments

Using the D4RL benchmark (Fu et al., 2021), we implemented two variants of the proposed method: **SWG** as described above, and **SWG-R**, which includes action resampling on top of our method. We considered the following methods as benchmarks:

- **Classical offline RL:** CQL (Kumar et al., 2020) and IQL (Kostrikov et al., 2022)
- **Diffusion-based resampling:** SfBC (Chen et al., 2023) and IDQL (Hansen-Estruch et al., 2023)
- **Diffusion-guided:** Diffuser (Janner et al., 2022), D-DICE (Mao et al., 2024), and QGPO (CEP) (Lu et al., 2023)
- **D-QL** (Wang et al., 2022), which trains a DM using Q-values as regularization term

Furthermore, our experiments are performed across four task groups:

1. **Locomotion:** the main offline RL benchmark, which features moving an agent across a plane.
2. **Ant Maze:** a more challenging task, where an ant agent is guided through a maze to reach a target.
3. **Franka Kitchen:** involves goal-conditioned manipulation in a realistic kitchen environment.
4. **Adroit Pen:** focuses on high-dimensional dexterous manipulation to reorient a pen.

Table 2: SWG against the state-of-art on the D4RL benchmark: Mean and standard deviation of the normalized returns (Fu et al., 2021) under 5 random seeds. Values within 5% of the best performing method in each task (row) are in bold font. SWG and SWG-R correspond to the standard and resampling version of our method, respectively. We used acronyms ME: Medium-Expert, M: Medium, MR: Medium-Replay, Avg: Average, HC: HalfCheetah, Hopp: Hopper, W2D: Walker2D, Kitchen: Franka Kitchen, Pen: Adroit Pen. See Appendix B.2 for full experimental details.

| Dataset | Env | CQL | IQL | SfBC | IDQL | Diffuser | D-QL | D-DICE | QGPO | SWG | SWG-R |
|---------|-----|-----|-----|------|------|----------|------|--------|------|-----|-------|
| ME | HC | 62.4 | 86.7 | **92.6** | **95.9** | 79.8 | **96.8** | **97.3** | 93.5 | **92.8 ± 0.6** | **93.5 ± 0.5** |
| ME | Hopp | 98.7 | 91.5 | **108.6** | **108.6** | **107.2** | **111.1** | **112.2** | 108.0 | **109.9 ± 3.2** | **111.0 ± 0.2** |
| ME | W2D | **111.0** | **109.6** | **109.8** | **112.7** | 108.4 | **110.1** | **114.1** | **110.7** | **110.7 ± 1.7** | **112.5 ± 0.3** |
| M | HC | 44.4 | 47.4 | 45.9 | 51.0 | 44.2 | 51.1 | **60.0** | 54.1 | 46.9 ± 0.2 | 46.9 ± 0.2 |
| M | Hopp | 58.0 | 66.3 | 57.1 | 65.4 | 58.5 | 90.5 | **100.2** | 98.0 | 68.6 ± 4.4 | 74.6 ± 5.2 |
| M | W2D | 79.2 | 78.3 | 77.9 | 82.5 | 79.7 | **87.0** | 89.3 | 86.0 | 74.0 ± 2.5 | 74.8 ± 2.1 |
| MR | HC | 46.2 | 44.2 | 37.1 | 45.9 | 42.2 | **47.8** | **49.2** | **47.6** | 42.5 ± 1.5 | 42.5 ± 1.5 |
| MR | Hopp | 48.6 | 94.7 | 86.2 | 92.1 | 96.8 | **101.3** | **102.3** | 96.9 | 73.9 ± 2.3 | 73.9 ± 2.3 |
| MR | W2D | 26.7 | 73.9 | 65.1 | 85.1 | 61.2 | **95.5** | 90.8 | 84.4 | 60.6 ± 4.1 | 79.9 ± 4.5 |
| **Avg (Loc)** | | 63.9 | 76.9 | 75.6 | 82.1 | 75.3 | **88.0** | **90.6** | 86.6 | 75.5 | 78.8 |
| Default | Umaze | 74.0 | 87.5 | 92.0 | **94.0** | - | **93.4** | **98.1** | 96.4 | 92.4 ± 2.6 | **98.3 ± 1.7** |
| Diverse | Umaze | **84.0** | 62.2 | **85.3** | 80.2 | - | 66.2 | **82.0** | 74.4 | 70 ± 4.2 | 71.7 ± 8.3 |
| Play | Med | 61.2 | 71.2 | 81.3 | 84.5 | - | 76.6 | **91.3** | 83.6 | **89.6 ± 2.3** | **90.0 ± 2.9** |
| Diverse | Med | 53.7 | 70.0 | 82.0 | 84.8 | - | 78.6 | 85.7 | 83.8 | **88.2 ± 5.2** | **91.7 ± 1.7** |
| Play | Large | 15.8 | 39.6 | 59.3 | 63.5 | - | 46.4 | **68.6** | **66.6** | 55.4 ± 4.6 | **66.7 ± 3.3** |
| Diverse | Large | 14.9 | 47.5 | 45.5 | 67.9 | - | 56.6 | **72.0** | 64.8 | 55.2 ± 3 | 63.3 ± 8.3 |
| **Avg (AntMaze)** | | 50.6 | 63.0 | 74.2 | **79.1** | - | 69.6 | **83** | 78.3 | 75.1 | **80.3** |
| Human | Pen | 37.5 | 71.5 | - | 76 | - | 72.8 | 84.4 | 72.8 | 78.5 ± 6.3 | **103.0 ± 5.0** |
| Cloned | Pen | 39.2 | 37.3 | - | 64 | - | 57.3 | 83.8 | 54.2 | 62.3 ± 6.7 | **91.7 ± 10.5** |
| **Avg (Adroit Pen)** | | 38.4 | 54.4 | - | 70 | - | 65.1 | 84.1 | 63.5 | 70.4 | **97.4** |
| Partial | Kitchen | 49.8 | 46.3 | 47.9 | - | - | 60.5 | **78.3** | - | 69.0 ± 4.9 | 73.3 ± 1.4 |
| Mixed | Kitchen | 51.0 | 51.0 | 45.4 | - | - | 62.6 | 67.8 | - | 61.0 ± 2.2 | **65.0 ± 0.8** |
| Complete | Kitchen | 43.8 | 62.5 | 77.9 | - | - | **84.0** | - | - | 72.3 ± 2.6 | 72.3 ± 2.6 |
| **Avg (Kitchen)** | | 48.2 | 53.3 | 57.1 | - | - | **69.0** | - | - | **67.4** | **70.2** |

These experiments were conducted with the expectile weight proposed in IDQL (Hansen-Estruch et al., 2023) modified as detailed in Sec. 5, and using $K = 15$ diffusion steps. See Appendix B.2 for more details.

The results shown in Table 2 reveal that the resampled version of our proposal SWG-R provided competitive performance over state-of-the-art DMs in the challenging environment tasks from Franka Kitchen and Adroit Pen. In Ant Maze, SWG-R also reported very competitive figures, outperforming all methods except D-DICE. However, in locomotion tasks, both SWG and SWG-R achieved average performance compared to previous methods. Also, note that SWG outperformed classical offline RL methods such as CQL and IQL in most tasks.

Based on the D4RL results, we argue that SWG-R offers a tradeoff between having a simpler training setup with fewer components than current DM guidance-based approaches at the cost of a lower performance with respect to the top-performing method, D-DICE (Mao et al., 2024), for the particular case of the Locomotion dataset. Moreover, incorporating resampling was pivotal for the enhanced performance of SWG-R over SWG. This addition strengthened the effectiveness of our framework, similarly to previous approaches. However, it is worth noting that the performance gain of SWG-R comes at the cost of an expanded hyperparameter search space during inference (see Appendix B.2). Additionally, we remark that the resampling was performed using the separate weight model $w$ (i.e., expectile weights).

Table 3: Ablation study for different weights models implemented alongside SWG on the locomotion D4RL benchmark.

|  | Expectile | Exp-expectile | Linex |
|---|---|---|---|
| Average (locomotion) | **75.5** | 69.9 | 63.6 |

## 6.3 Ablation study for different weights

We then evaluated SWG's performance under different weight formulations in Sec. 5. This test was conducted on D4RL locomotion, using the same model architecture and hyperparameter tuning to ensure a fair comparison. Furthermore, we also tuned the guidance scale in each case to find the best possible performance. The weight formulations were implemented as follows:

- **Expectile weight**: Learnt with the same setup adopted in the main results, described in Sec. 5.

- **Exponential expectile weight**: We used the same setup as for IDQL expectile weights, and considered a temperature $\beta = 3$ to exponentiate the advantage function $e^{\beta A(s, \mathbf{a})}$. To ensure stability during training, we clipped the exponential weights to $(0, 80]$.

- **Linex weight**: We followed IDQL (Hansen-Estruch et al., 2023), and used $\alpha = 1$.

The results, shown in Table 3, reveal that expectile yielded the best performance, with the exponential expectile showing moderate performance, and lastly Linex performing substantially worse. We conjecture that this is due to the training instability introduced by the exponential weights, in general. Most importantly, these results are consistent with the findings reported in IDQL (Hansen-Estruch et al., 2023).

## 6.4 Ablation over guidance scale sensitivity

We also studied the effect of using a fixed guidance scale across all tasks, without tuning $\rho$ for each task individually. Recall that the choice of the guidance scale $\rho$ controls the strength of the guidance score, which translates into a sharper target distribution concentrated around high-value weights. This effect can be seen in the modified noise prediction $\hat{\epsilon} := \epsilon_\theta(\mathbf{z}_k, k) - \rho \sigma_k \nabla_{\mathbf{z}_k} \log \left(\phi_w\left(\frac{\mathbf{z}_k - \sigma_k \epsilon_\theta(\mathbf{z}_k, k)}{\alpha_k}\right)\right) = \epsilon_\theta(\mathbf{z}_k, k) - \sigma_k \nabla_{\mathbf{z}_k} \log \left(\phi_w\left(\left(\frac{\mathbf{z}_k - \sigma_k \epsilon_\theta(\mathbf{z}_k, k)}{\alpha_k}\right)^\rho\right)\right)$. We conduct these experiments using expectile weights, which, as shown in Sec. 6.3, yield superior performance.

Overall, from Tables 4 and 5, we observe that the best performance was generally achieved with $\rho = 10$ or $\rho = 15$. As expected, using low guidance scales ($\rho < 5$) resulted in suboptimal performance, as the guidance signal was not sufficiently strong. On the other hand, employing large guidance scales ($\rho > 20$) led to poorer performance, since an overly strong guidance signal tends to produce out-of-distribution samples.

Table 4: Guidance scale $\rho$ sensitivity in Locomotion tasks.

| $\rho$ | 1 | 5 | 10 | 15 | 20 | 25 | 30 |
|---|---|---|---|---|---|---|---|
| Average | 63.8 | 72.0 | 72.9 | 73.1 | 71.2 | 69.2 | 68.8 |

Table 5: Guidance scale $\rho$ sensitivity in Ant Maze tasks.

| $\rho$ | 1 | 5 | 10 | 15 | 20 | 25 | 30 |
|---|---|---|---|---|---|---|---|
| Average | 24.6 | 37.9 | 69.6 | 68.0 | 67.3 | 61.8 | 57.8 |

### 6.5   How does SWG's inference time scale with its size?

Since SWG requires computing gradients through DM, we investigated its scalability properties through two experiments examining both depth and width scaling of its networks. For depth scaling, we used a simple multilayer perceptron (MLP) with a hidden dimension of 2056 and measured the inference time across varying numbers of layers. For width scaling, we used a fixed 4-layer MLP with varying hidden dimension, measuring the average inference time accordingly (see the Appendix B.3).

As shown in Figures 4 and 5, SWG exhibited a linear scaling with respect to model depth and exponential scaling with respect to the width scaling (hidden dimension). Additionally, and as expected, sampling with guidance (i.e., computing gradients) translates into larger inference times. However, qualitatively, we can observe that both sampling methods scale at a similar rate. Note that scaling issues related to gradient computation can be easily avoided by designing the DM to output the weight component in its early layers. This allows gradients to be computed with respect to a fixed module, without requiring backpropagation through the entire model. Consequently, the additional computational overhead of SWG remains constant, even as the sub-component of the model responsible for action generation increases in scale.

The training time remains essentially the same as that of the related methods (Hansen-Estruch et al., 2023; Mao et al., 2024) when using a similar architecture. The SWG training time measurements are reported in the Appendix B.4.

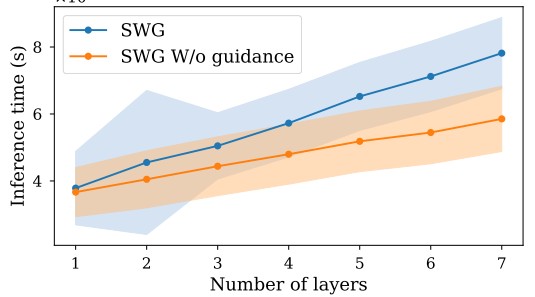

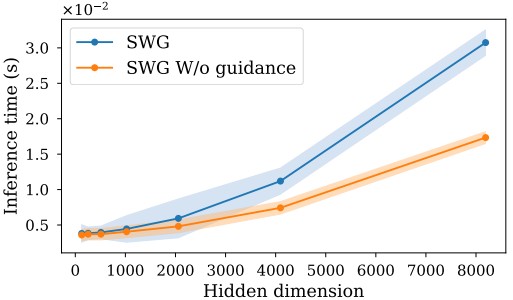

Figure 4: SWG depth scaling: inference time (s) vs number of layers (mean $\pm$ 2 standard deviations over 10,000 runs).

Figure 5: SWG width scaling: inference time (s) vs hidden dimension (mean $\pm$ 2 standard deviations over 10,000 runs).

## 7   Conclusion

We have proposed a joint DM over actions and weights for optimal policy discovery in offline RL. Unlike current DM-based methods, our approach avoids the computation of a guidance score network. In our proposal, the computation of the scores is straightforward, as they are readily available from the DM implementation. The proposed method, termed *Self-Weighted Guidance* (SWG), owes its name to having the weight guidance information embedded into the DM. Compared to previous methods, SWG conducts exact guidance using a streamlined setup which requires a single DM, and provides novel insights arising from modeling weights as random variables, thus demonstrating one more time the flexible modeling capabilities of DM's.

Via toy examples and the standard D4RL benchmark, we have provided experimental validation of SWG's ability to: i) effectively produce samples of the target distribution, and ii) perform on par with —or better than— the state-of-the-art DMs on challenging environments such as Ant Maze, Franka Kitchen and Adroit Pen, while having a simpler training setup. We have also validated the ability of SWG to leverage resampling for improved performance. However, we observe that, in general, SWG performs sub optimally on mixed-action datasets (i.e., Mixed, Medium-Replay, Diverse), which is likely due to either the choice of weight function and/or the architecture of the DM, both of which may limit how effectively the joint action–weight

distribution is learned. Furthermore, we have provided an ablation study for different weight formulations within SWG, and also studied how SWG's inference time scales with its network's width and depth. We stress that the main advantage of SWG comes from its simplicity and reduced training setup compared to previous guidance methods.

Lastly, we claim that the proposed DM-based sampling methodology, which is guided by the same DM, has the potential for applications beyond the offline RL setting. Tasks involving weighted distributions, such as controllable image and text generation (Dhariwal & Nichol, 2021; Li et al., 2022), can be addressed by the proposed SWG approach.

**Limitations and future work.** We have provided a proof of concept for SWG, which performed among the top 2 methods across D4RL's datasets except in Locomotion, where SWG was outperformed by QGPO, D-QL and D-DICE. To address intrinsic limitations, we note that jointly learning actions and weights reduces flexibility (changing the guidance requires retraining the entire diffusion model) and makes training more challenging compared to learning actions alone, potentially needing higher-capacity networks. Both issues could be mitigated by dedicating a portion of the network exclusively to learning weights. Future work could further improve SWG with tailored weight models and more sophisticated ODE solvers.

**Broader impacts.** This work presents a promising methodology for offline RL, where policies can be learned from a controlled offline dataset. However, as with most diffusion models, potential misuse includes the generation of harmful data, such as fake images, videos, or audio.

**Acknowledgements** JRdS acknowledges financial support from ANID-Chile grants Basal CIA250010 and Fondecyt-Regular 1251823.

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

# A  Proofs and derivations

## A.1  Proof of Proposition 3.3: computation of the target score

We follow the derivation utilized in (Lu et al., 2023).

Let us express the normalizing constant of $p_0(\mathbf{z}_0)$ as

$$A = \int q_0(\mathbf{z}_0)\phi_w(\mathbf{z}_0)d\mathbf{z}_0 = \mathbb{E}_{q_0(\mathbf{z}_0)}\left[\phi_w(\mathbf{z}_0)\right].$$

Recall that $\phi_w(\cdot)$ *extracts* the $w$ component of $z$. Then, we have:

$$p_0(\mathbf{z}_0) = \frac{q_0(\mathbf{z}_0)\phi_w(\mathbf{z}_0)}{A}.$$

According to the definition of diffusion models, we have:

$$p_k(\mathbf{z}_k) = \int p_{k0}(\mathbf{z}_k|\mathbf{z}_0)p_0(\mathbf{z}_0)d\mathbf{z}_0 = \int p_{k0}(\mathbf{z}_k|\mathbf{z}_0)q_0(\mathbf{z}_0)\frac{\phi_w(\mathbf{z}_0)}{A}d\mathbf{z}_0 = \frac{1}{A}\int q_{k0}(\mathbf{z}_k|\mathbf{z}_0)q_0(\mathbf{z}_0)\phi_w(\mathbf{z}_0)d\mathbf{z}_0$$

$$= q_k(\mathbf{z}_k)\frac{1}{A}\int q_0(\mathbf{z}_0|\mathbf{z}_k)\phi_w(\mathbf{z}_0)d\mathbf{z}_0 = q_k(\mathbf{z}_k)\frac{\mathbb{E}_{q_0(\mathbf{z}_0|\mathbf{z}_k)}\left[\phi_w(\mathbf{z}_0)\right]}{A},$$

and finally

$$\nabla_{\mathbf{z}_k}\log p_k(\mathbf{z}_k) = \nabla_{\mathbf{z}_k}\log q_k(\mathbf{z}_k) + \nabla_{\mathbf{z}_k}\log \mathbb{E}_{q_0(\mathbf{z}_0|\mathbf{z}_k)}\left[\phi_w(\mathbf{z}_0)\right].$$

## A.2  Proof of Theorem 3.5

Recall that we want to sample from $p(\mathbf{z})$ in equation 10.

Let us assume that the weight function $w(\cdot)$ is known, so we have access to a labeled dataset consisting of samples $\mathbf{z}_0 = (\mathbf{a}_0, w(\mathbf{a}_0))$. We suppose a $K$ step diffusion process over $\mathbf{z}_0$, as $\mathbf{z}_k = \alpha_k\mathbf{z}_0 + \sigma_k\epsilon_0$, $\epsilon \sim \mathcal{N}(0,\mathbf{I})$.

To address the problem in Equation equation 4, let us assume that we train a noise-predicting diffusion model on the data $\mathbf{z}_0$, $\epsilon_\theta(\mathbf{z}_k, k)$, $k \in [0, K]$, parametrized by $\theta$. Then, the optimal diffusion model $\epsilon^*$ is defined by the following optimization problem.

$$\operatorname*{argmin}_\theta ||\epsilon_0 - \epsilon_\theta(\mathbf{z}_k, k)||_2^2. \tag{17}$$

Let us also recall the data prediction formula (Efron, 2011; Kingma et al., 2021) for diffusion models, which states that the optimal diffusion model satisfies:

$$\mathbb{E}_{q_{0k}(\mathbf{z}_0|\mathbf{z}_k)}[\mathbf{z}_0] \approx z_\theta^*(\mathbf{z}_k, k) := \frac{\mathbf{z}_k - \sigma_k\epsilon_\theta^*(\mathbf{z}_k, k)}{\alpha_k}. \tag{18}$$

We can use this formula to compute the intractable score in equation 10. Since the extraction function $\phi_w(\cdot)$ is linear (see Def. 3.1 in Sec. 3), we can swap the expectation and $\phi_w$ to obtain

$$\mathbb{E}_{q_{0k}(\mathbf{z}_0|\mathbf{z}_k)}[\phi_w(\mathbf{z}_0)] = \phi_w(\mathbb{E}_{q_{0k}(\mathbf{z}_0|\mathbf{z}_k)}[\mathbf{z}_0]) = \phi_w\left(\frac{\mathbf{z}_k - \sigma_k\epsilon_\theta^*(\mathbf{z}_k, k)}{\alpha_k}\right). \tag{19}$$

Finally, plugging this term in equation 10

$$\nabla_{\mathbf{z}_k}\log p_k(\mathbf{z}_k) = \underbrace{\nabla_{\mathbf{z}_k}\log q_k(\mathbf{z}_k)}_{\frac{-\epsilon_\theta^*(\mathbf{z}_k, k)}{\sigma_k}} + \underbrace{\nabla_{\mathbf{z}_k}\log\left(\phi_w\left(\frac{\mathbf{z}_k - \sigma_k\epsilon_\theta^*(\mathbf{z}_k, k)}{\alpha_k}\right)\right)}_{\text{self guidance}}. \tag{20}$$

Note that if we utilize a data prediction diffusion model $\mathbb{E}_{q_{0k}(\mathbf{z}_0|\mathbf{z}_k)}[\mathbf{z}_0] = z_\theta^*(\mathbf{z}_k, k)$ the target score can be expressed as:

$$\nabla_{\mathbf{z}_k} \log p_k(\mathbf{z}_k) = \underbrace{\nabla_{\mathbf{z}_k} \log q_k(\mathbf{z}_k)}_{\frac{-\epsilon_\theta^*(\mathbf{z}_k,k)}{\sigma_k}} + \underbrace{\nabla_{\mathbf{z}_k} \log\left(\phi_w(z_\theta^*(\mathbf{z}_k, k))\right)}_{\text{self guidance}}. \tag{21}$$

Notice we must make the assumption that $\phi_w(\frac{\mathbf{z}_k - \sigma_k \epsilon_\theta^*(\mathbf{z}_k,k)}{\alpha_k}) > 0$, which the optimal diffusion model satisfies if the weight function is positive $w(\mathbf{a}) > 0 \ \forall \mathbf{a} \in supp(A)$.

## B    Experimental details

### B.1    2-D Toy examples

We followed the toy examples proposed in CEP (Lu et al., 2023), predefining a weight function $w(\mathbf{a})$ and a dataset distribution $q(\mathbf{a})$. We built the dataset $\mathcal{D}$ with 1M samples from $q(\mathbf{a})$ and extended it to $\mathcal{D}_Z = \{\mathbf{z} = (\mathbf{a}, w(\mathbf{a}))\}_{i=1}^N$ using the predefined weight function.

We then trained a diffusion model $\epsilon_\theta$ on the extended dataset $\mathcal{D}_Z$. For all toy examples, the diffusion model was trained for $K = 100$ diffusion steps, using a Cosine noise schedule (Nichol & Dhariwal, 2021). The architecture consisted of a 6-layer Multi-Layer-Perceptron (MLP) with hidden dimension 512 and GeLU activations. We trained over $200k$ gradient steps and batch size 1024 using the Adam optimizer (Kingma & Ba, 2015) with learning rate $10^{-4}$.

To avoid training a separate diffusion model for each $\beta$, we conditioned the diffusion model on $\beta$, $\epsilon_\theta(\mathbf{z}_k, k, \beta)$, allowing us to train a single model per distribution $q(\mathbf{a})$. We trained over $\beta \in [0, 20]$. For the sampling process, we set the guidance scale $\rho$ to 1, as $\rho = 1$ recovers the unbiased estimate of the target distribution score.

Figure 6 presents four additional experiments on the swiss roll, rings, moons, and checkerboard distributions.

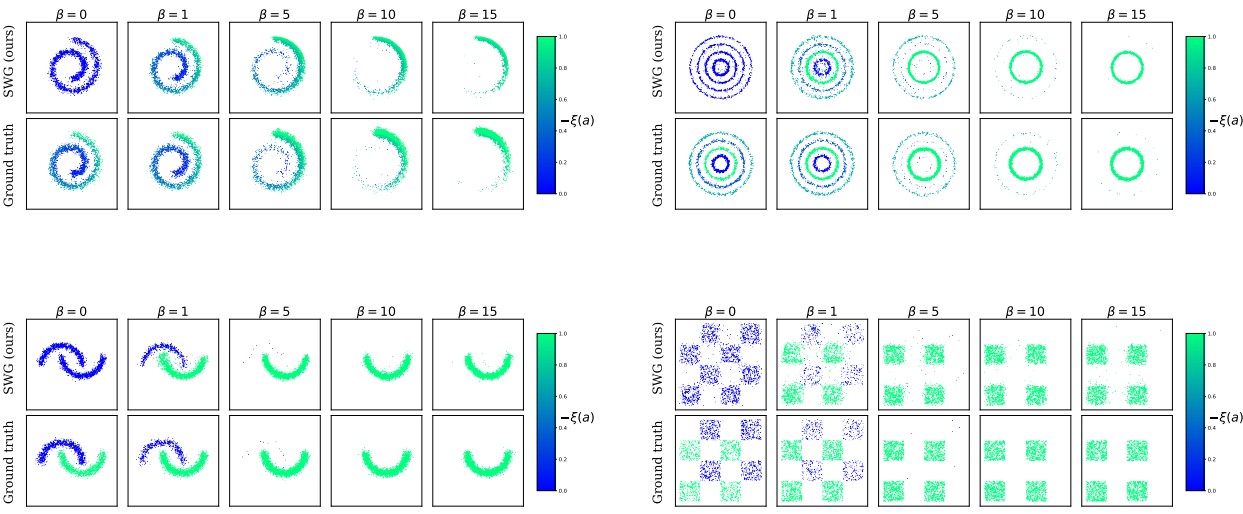

Figure 6: Additional examples for the experiment in Sec. 6.1. Target distributions from top-left, clockwise: swiss roll, rings, moons, and checkerboard distributions

### B.2    D4RL experiments

For the D4RL Locomotion tasks, we averaged mean returns over 5 random seeds and 20 independent evaluation trajectories per seed. For Ant Maze tasks, we averaged over 50 evaluation trajectories and five random seeds.

The results found in Table 2 correspond to the normalized score in each domain, following the suggestion of the authors of the dataset (Fu et al., 2021). The results of our method correspond to the evaluation in the last training step. The results for other methods are taken from their respective original articles, except for Franka Kitchen, QGPO and IDQL results were obtained from (Li, 2024) and (Park et al., 2025), respectively.

Before training the models, we followed the suggestions found in (Fu et al., 2021) and standardized locomotion task rewards in the datasets by dividing them by the difference of returns of the best and worst trajectories in each dataset. For the Ant Maze dataset, we subtracted 1 from the rewards.

To perform the offline RL experiments, it was first necessary to estimate the weight function. We used the expectile weights presented in IDQL (Hansen-Estruch et al., 2023) for the main D4RL results, as we found that they produce the best performance compared to the weights tested.

For the expectile weights, we needed to estimate the value functions $Q(s, \mathbf{a})$ and $V(s)$. We followed the same setup indicated in IQL (Kostrikov et al., 2022) and IDQL (Hansen-Estruch et al., 2023), which consists of estimating Q-values $Q_\psi$ and the value function $V_\phi$, both with a 2-layer MLP with ReLU activations and 256 hidden units. We trained the networks with Adam optimizer (Kingma & Ba, 2015) for 1M gradient steps, using learning rate $3e-4$ and batch size 256. We used double Q-learning (Fujimoto et al., 2018) and soft updates (Lillicrap et al., 2016) to stabilize the training. For the soft update, we used 0.005 in all experiments to update the Q-network. We also set $\gamma = 0.99$ for all tasks, and $\tau = 0.7$ for the expectile loss, except for Ant Maze where we use $\tau = 0.9$. The training procedure is described in Algorithm 3.

---

**Algorithm 3** Expectile Loss critic learning

**Input:** Dataset $\mathcal{D} = \{(s_i, \mathbf{a}_i, r_i, s'_i)\}_{i=1}^N$, discount factor $\gamma$, expectile $\tau$, batch size $B$, learning rate $\alpha$, target update parameter $\lambda$.
**Initialize:** Q-network $Q_\psi$, target Q-network $Q_{\hat\psi}$, value network $V_\phi$.
**for** each gradient step **do**
    Get $B$ samples from $\mathcal{D}$
    Compute expectile loss $L_V(\phi) = \mathbb{E}_{(s,\mathbf{a})\sim B}\left[L_\tau^2\left(Q_{\hat\psi}(s,\mathbf{a}) - V_\phi(s)\right)\right]$
    Update value network $\phi \leftarrow \phi - \alpha\nabla_\phi L_V(\phi)$
    Compute Q-value loss $L_Q(\psi) = \mathbb{E}_{(r,s,\mathbf{a},s')\sim B}\left[(r + \gamma V_\phi(s') - Q_\psi(s,\mathbf{a}))^2\right]$
    Update Q-network $\psi \leftarrow \psi - \alpha\nabla_\psi L_Q(\psi)$
    Update target Q-network $\hat\psi \leftarrow (1-\lambda)\hat\psi + \lambda\psi$
**end for**
**Return:** Optimized Q-network $Q_\psi$ and value network $V_\phi$

---

Once the value functions are learned, we were able to calculate the weights as detailed in Sec. 5. Then, we extended our dataset to $\mathcal{D}_Z$ following the methodology described in Sec. 3. We trained a diffusion model $\epsilon_\theta$ on this extended dataset for $K = 15$ diffusion steps in all experiments, using Adam optimizer (Kingma & Ba, 2015) with learning rate $3e-4$ for all tasks, except for Adroit Pen where we use $3e-5$. We used a batch size of 1024, a variance-preserving noise schedule (Song et al., 2021), and trained for 1M gradient steps in all tasks, except for Ant Maze, where we used 3M gradient steps.(Janner et al., 2022; Hansen-Estruch et al., 2023). The diffusion model consisted of a ResNet of 3 Residual blocks, with hidden dim 256 and Layer Normalization (Ba et al., 2016). We used a dropout (Srivastava et al., 2014) of 0.1 in all tasks. For the weight component output, we used the features from the middle layer of the ResNet and passed them through a simple MLP with a hidden dimension of 256 to obtain a more expressive representation of the weights. This architecture closely follows the one used in IDQL (Hansen-Estruch et al., 2023), with the exception of the additional MLP dedicated to the weight component. However, this MLP did not introduce a significant number of additional parameters. The complete ResNet architecture contained approximately 1.8M parameters, ensuring a fair comparison in model capacity, as other guidance methods (Lu et al., 2023; Mao et al., 2024) use U-Net architectures (Ronneberger et al., 2015) with a comparable number of total parameters.

During inference, we tuned the guidance scale for each task. We swept over $\rho \in \{1, 5, 10, 15, 20, 25, 30\}$, such relatively large guidance scales were considered due to the expectile weights used, which were bounded in

range. The guidance scale used for each experiment to obtain SWG results can be found in Tables 6, 7, 8 and 9. The results found in Table 2 correspond to the evaluation of the model at the last training step. For SWG-R (with the added resampling step), we conducted an additional hyperparameter search over the batch size, sweeping $M \in \{1, 2, 4, 8, 16\}$.

Table 6: Utilized guidance scales for locomotion results in Table 2

|  | Half Cheetah | Walker2d | Hopper |
|---|---|---|---|
| Medium-Expert | 15 | 5 | 5 |
| Medium | 20 | 5 | 20 |
| Medium-Replay | 30 | 10 | 10 |

Table 7: Utilized guidance scales for Ant Maze results in Table 2

|  | U-Maze | Medium | Large |
|---|---|---|---|
| Play/Default | 5 | 15 | 10 |
| Diverse | 1 | 15 | 10 |

Table 8: Utilized guidance scales for Franka Kitchen results in Table 2

| Partial | Mixed | Complete |
|---|---|---|
| 10 | 15 | 10 |

Table 9: Utilized guidance scales for Adroit Pen results in Table 2

| Human | Cloned |
|---|---|
| 1 | 25 |

### B.3 Scaling of inference time in SWG

Each data point in Figures 4 and 5 represents the average inference time with our method for $K = 15$ diffusion steps, obtained over 10,000 runs of the algorithm, along with the corresponding error region. For completeness, we report the total number of parameters in each network used in our experiments. The table below shows the parameter counts corresponding to different width and depth configurations:

Table 10: Number of parameters for each network configuration used in our experiments. Left: scaling with width (hidden size). Right: scaling with depth (number of layers).

| Width (hidden size) | Parameters | | Depth (layers) | Parameters |
|---|---|---|---|---|
| 128 | 180K | | 1 | 660K |
| 256 | 370K | | 2 | 4.5M |
| 512 | 1M | | 3 | 9.1M |
| 1024 | 3.5M | | 4 | 13.3M |
| 2056 | 13.3M | | 5 | 17.5M |
| 4096 | 51.5M | | 6 | 21.8M |
| 8192 | 203.7M | | 7 | 26M |

### B.4 Computational resources

Our implementation is based on the jaxrl repository (Kostrikov, 2021) using the JAX (Bradbury et al., 2018) and Flax (Heek et al., 2024) libraries. We also provide an implementation of our method in PyTorch (Paszke et al., 2019). All experiments were conducted on a 12GB NVIDIA RTX 3080 Ti GPU. Training takes approximately 0.5 hours for value function learning (i.e., weight training, 1M steps), and an additional 0.5 hours for 1M steps of the diffusion model. However, since experiments can be run in parallel, the overall training time can be significantly reduced. The code used for training and experimentation can be found in the `SWG repository`

