# OpenReview forum: "Diffusion Self-Weighted Guidance for Offline Reinforcement Learning"
_TMLR — Accepted by TMLR_

### Review · Reviewer_Su2U · 2025-09-22

**Summary Of Contributions:**

**Problem Setup:** Offline RL addresses scenarios where agents must learn effective policies from static datasets without environment interaction. The main challenge is avoiding dependency on out-of-distribution action evaluation when learning the target policy, as inferring dynamics or rewards for unseen state-action pairs typically leads to overestimation errors. Beyond model-based methods or trajectory modeling approaches, most methods model the target policy as π(a|s) ∝ μ(a|s)w(s,a), where μ is the behavior policy and w represents action quality. Methods like IQL learn these weights using advantage functions A(s,a) = Q(s,a) - V(s). Here the value are calculated by leveraging expectile regression, which avoids querying good actions by learning expectiles of behavioral Q-functions rather than requiring max/min operations over the action space.

In the diffusion paradigm, models can effectively capture behavioral distributions, but extracting improved policies requires additional mechanisms. Resampling approaches like IDQL sample multiple candidates from the behavioral diffusion model and select actions using learned critics. Guidance-based methods like QGPO learn explicit guidance networks to steer the diffusion process toward higher-value actions during sampling. D-DICE combines both paradigms—using in-sample guidance learning to generate high-quality candidates, then selecting among them—achieving state-of-the-art performance across D4RL benchmarks.


**Proposed Solution:** The proposed SWG approach attempts to simplify this pipeline by jointly modeling actions and weights within a single diffusion model. Instead of learning separate models for behavioral policy and guidance networks, SWG trains on augmented data tuples (state, action, weight) where weights come from learned advantage functions. During sampling, at each denoising step, the guidance mechanism uses the data prediction formula to estimate what the clean (action, weight) pair should be, then extracts the weight component from this prediction and computes its gradient ∇_{z_k} log(ϕ_w(z₀_predicted)). This gradient is used to modify the noise prediction, steering the denoising process toward configurations that would yield higher weights. This eliminates the need for separate guidance networks since the weight predictions come directly from the same diffusion model, making the guidance signal a natural byproduct of the joint modeling approach.


**Strengths:**
- **Direct Score Computation:** The joint modeling of actions and weights enables direct computation of guidance scores from the diffusion model's own prediction head, eliminating the need to approximate intractable expectations E[w(a_0)|a_k] through auxiliary networks with complex loss functions.
- **Simplified Training Pipeline:** Uses only standard diffusion loss minimization (MSE on predicted noise), avoiding slightly complex exponential terms, contrastive objectives, and potential training instabilities found in D-DICE and CEP guidance networks. This consolidates behavioral modeling and guidance into a single network, reducing the number of components that need to be trained, tuned, and deployed.
- **Good set of experiments and Ablations:**  Includes comprehensive ablations on guidance scaling, inference time scaling with model size (with/without guidance), and different weight formulations. Demonstrates competitive performance with established baselines including IQL and D-QL across D4RL benchmarks.

**Weakness:**
* **Fundamental Performance Gap Without Justification:** SWG consistently across all tasks underperforms state-of-the-art methods, with D-DICE (case in point D-DICE achieves 90.6 vs SWG's 75.5 on locomotion tasks). This substantial gap undermines the paper's central premise, especially since D-DICE already effectively solves the "intractable" guidance problem that SWG claims to address. So some questions still arise.
    * Why joint modeling should outperform specialized guidance networks ? Should they ?
    * How do we quantify practical benefits the claimed simplicity actually provides ?  Why should one accept the performance degradation when there is no evidence that this framework can be better tuned for a specific problem we care about ?
    * Why did D-DICE actually perform better ? Is it that specialized guidance is just better and modeling guidance signals ?
    * How much of performance of D-DICE is attributed to better guidance modeling as compared to the resampling of actions ?
    * Can the performance gap be mitigated by adding resampling on top of SWG? How does resampling budget affect performance?

**Audience:**

Yes

**Audience Explanation:**

The paper addresses an important problem in offline RL and proposes a theoretically interesting approach.
However, the practical impact is not fully validated given the performance gaps with existing methods. Still The joint modeling perspective offers insights that could inspire future work.

**Claims And Evidence:**

No

**Claims Explanation:**

**Supported Claim:** The paper's mathematical derivations are technically sound, and the toy experiments provide clear evidence that SWG can sample from target distributions exactly when the weight function is known.

**Unsupported Claims:**

- While the D4RL experimental evaluation is comprehensive, it consistently shows SWG underperforming state-of-the-art methods across most benchmarks. The authors present this as a reasonable trade-off for simplicity, but provide insufficient evidence that this trade-off is worthwhile.

- The claim that SWG offers a "streamlined training pipeline" is supported by the elimination of separate guidance networks, but the practical benefits remain unclear given that weight function learning is still required. The paper subtly assumes architectural elegance translates to practical value, which may be true but needs a more thorough investigation.

**Requested Changes:**

Proposed Critical Changes for Acceptance:
- **Strengthen the motivational argument:** Either empirically demonstrate clear practical advantages (training stability, computational efficiency, memory usage) that justify the performance trade-off, or reframe the contribution as a theoretical exploration rather than a practical improvement.
- **Decompose D-DICE Contributions:** Isolate how much of D-DICE's performance stems from guidance quality vs. action resampling through ablations with D-DICE variants (guidance-only, selection-only).
- **Hybrid Approach Evaluation:** Test whether adding resampling on top of SWG can close the performance gap, and analyze how resampling budget affects results.

---

> ### Author Response · Authors · 2025-10-24
>
> Dear Reviewer,
>
> We sincerely thank you for your valuable comments and for suggesting the incorporation of a resampling step into SWG. This addition substantially improved our results, making them more competitive with state-of-the-art methods. We have carefully addressed all your comments and updated the paper accordingly (changes are highlighted in blue).
>
> Regarding the weaknesses you identified, we have the following comments/answers:
>
> *Joint modelling.* We understand that the joint modeling approach does not necessarily improve upon a specialized guidance network. Conversely, our goal is to validate the idea that a "simpler approach" could theoretically provide exact guidance, similar to a specialized guidance network. Here, a "simpler approach" refers to a joint DM-based model that uses the standard DM loss, as opposed to training the DM on actions and training an additional specialized guidance network with a loss defined separately.
>
> *Quantification of practical benefits.* One way to quantify the practical benefits directly is through performance and computational complexity, such as training and inference times. In our work, however, we only aimed to provide a proof of concept for the SWG approach and not a fully developed, competitive SWG, which would require more powerful solvers, tailored weight functions, and, in general, a more ambitious architecture. We plan to address these issues in our future work.
>
> *Why D-DICE performed better.* D-DICE is the main benchmark for testing our method, and we used the results published in the original paper. We tried to replicate D-DICE, but could not achieve the same results. This may be due to particular implementation tweaks, the use of more powerful solvers, or the hyperparameters used for sampling. Therefore, we cannot provide an objective explanation for why D-DICE outperformed our proposal (SWG) in our initial submission. However, our updated results now include a resampling step, which improved our performance and outperformed D-DICE in some datasets.
>
> Regarding the requested changes:
>
> Motivational Argument: We acknowledge that quantifying the benefits of our method is challenging. Therefore, we emphasize our contribution as a theoretical exploration. Throughout the paper, mainly in the abstract, introduction, experiments, and conclusions, we clearly state that the main practical benefit is qualitative rather than quantitative (i.e., simpler training setup). We thank the reviewer for this suggestion.
>
> D-DICE: As mentioned earlier, we made significant efforts to reproduce the D-DICE results. However, despite adhering to the instructions in the paper and using the provided code, we encountered difficulties in achieving stable critic training. Consequently, performing additional ablation studies on this method was not possible.
>
> Hybrid Approach Evaluation: We greatly appreciate this suggestion. We now offer two versions of our SWG method: one without resampling and one with resampling (see Table 2). SWG with resampling closes the performance gap with previous methods and achieves top performance in the AntMaze, Adroit Pen, and Franka Kitchen environments. This improvement in performance comes at the cost of a broader hyperparameter search during inference.

---

### Review · Reviewer_Upva · 2025-10-04

**Summary Of Contributions:**

This work studies diffusion models (DMs) for offline reinforcement learning (RL), where the target policy weighs between the behavior policy and a critic. DM-based methods approximate the behavior policy with a DM, then learn the weight function/critic via resampling or using a guidance network. This latter step is challenging because of intractable score functions and entangled weights and actions that yield numerical instability. The proposed method, SWG (self-weighted guidance), avoids those issues by embedding the guidance network within the same DM used for sampling from the behavior policy. Specifically, SWG leverages the fact that weights are deterministic functions of actions to adapt the guidance loss accordingly.

**Audience:**

Yes

**Audience Explanation:**

Avoiding additional training for the weights could be beneficial and make DMs for offline RL more accessible, but the authors should develop on the loss incurred by such avoidance.

**Claims And Evidence:**

No

**Claims Explanation:**

As a disclaimer, I am unfamiliar with diffusion models and their training methods.
However, the problem description and solutions *lack clarity* and self-contained explanations:
- How does IDQL generalize IQL? How does Eq. (5) relate to Eq. (4) or the loss used for expectile regression?
- How does one solve the reverse diffusion process after Eq. (7)? What does the score function in (9) model, and where does its relation with the noise come from?

From a broader perspective, the *motivation* for using DMs in offline RL remains unaddressed in this work. Why privilege DM over other generative models, e.g., ensembles? The approach relies on weighted behavior policies: how limiting is this assumption? Does SWG specifically cope with out-of-data actions compared to other similar methods?

The *performance of SWG* compared with other baselines is not convincing. The authors claim that SWG competes with other DM-based methods, but D-DICE typically outperforms in every benchmark. The potential instabilities mentioned after Eq. (22) are not very convincing to me, and a more thorough comparison with those SOTA DM-based methods would have been much appreciated in addition to the ablation studies proposed in this work.

The *limitations* of SWG are not properly addressed. Those mentioned in Sec. 7 do not refer to shortcomings that are intrinsic to SWG but rather to incomplete parameter and computational cost optimizations. I would expect the authors to explore the limitations of removing the guidance network instead: there should be a trade-off. We arguably win in stability and cost but lose in performance/sth else?

**Requested Changes:**

Besides my substantive comments above, the following minor changes should be made:
- Remove equation numbering when not referred to in the text
- Equation referring should be in parentheses + capital letter (Equation (3) instead of equation 3)
- Table 1 is superfluous to me -- these characteristic comparisons are already given in the text
- Changes in notations are confusing -- from Sec. 2 to 3, $\pi$ becomes $p$, $\mu$ becomes $q$? Please double-check consistency to avoid misunderstanding.
- The first sentence after Eq. (9) does not mean anything. Please review phrasing here, as well as in other places (repetitive words, starting sentences with Though, etc.)

---

> ### Author Response · Authors · 2025-10-24
>
> Dear Reviewer,
>
> Thank you for the constructive review. We reply below and have updated the paper in light of your suggestions.
> ### Reply to questions
>
> - How IDQL generalises IQL: In IQL (Implicit Q-Learning) [2], expectile regression is used to learn Q and V, yielding two advantages: (i) avoiding out-of-sample queries, and (ii) promoting improved actions beyond the dataset. It is “implicit” as Q-values are defined via the loss, not explicit querying. IDQL [1] generalises this by allowing any convex implicit loss, showing such losses induce weighted policy distributions.
>
> - Reverse diffusion process: Training the diffusion model (Eq. 3) learns to denoise samples. Once trained, it transforms Gaussian noise into data through iterative denoising, effectively sampling from $\mathcal{N}(0,I)$ toward the target distribution. The diffusion algorithm is detailed in [3].
>
> - Relationship of noise and score function:
> From the forward process:
> $q_k(a_k|a_0)=\mathcal{N}(a_k;\alpha_k a_0,\sigma_k^2 I)$,
> the conditional score is
> $\nabla_{a_k}\log q_k(a_k|a_0)=-\frac{a_k-\alpha_k a_0}{\sigma_k^2}$.
> Since $a_k=\alpha_k a_0+\sigma_k\epsilon_0$, $\epsilon_0\sim\mathcal{N}(0,I)$, then:
> $\nabla_{a_k}\log q_k(a_k|a_0)=-\frac{1}{\sigma_k}\epsilon_0.$
> - Motivation for Using Diffusion Models in Offline RL:
> Our motivation stems from two factors:
> (1) Diffusion models (DMs) are more expressive than classical architectures (e.g., MLPs), capturing complex, multimodal action distributions [5].
> (2) Their score-based formulation allows flexible sampling from conditioned or weighted distributions through guidance. Unlike autoregressive transformers, DMs separate conditioning via an auxiliary guidance network that steers the generation process. In score-based models, sampling from product distributions corresponds to summing their log-gradients equation 4.
>
> - Assumption of weighted behaviour policies: Modeling offline RL with a weighted behaviour policy is not restrictive. As shown in [4], the optimal action distribution can be expressed as $\pi\propto\mu\exp{\beta Q(s,a)}$. This formulation is highly expressive, as the temperature parameter $\beta$ controls the weighting strength. Weighted behavior policies are common in offline RL [1]. In our work, we assume a general weighted behavior policy, which remains flexible since the weight function can be any real-valued function (see Section 3.1 after Eq. 6), without restrictive assumptions.
>
> - SWG and out-of-sample actions:
> SWG, like IDQL and IQL [1, 2], avoids out-of-sample queries during training due to the diffusion loss. At inference, however, it samples from the weighted behaviour policy, where out-of-distribution actions may appear.
>
> - Performance:
> Following Reviewer Su2U’s suggestion, we introduced resampling on top of SWG, yielding notable improvements. As shown in Table 2, SWG + resampling closes the gap with prior methods, achieving top results in AntMaze, Adroit Pen, and Franka Kitchen.
>
> - Intrinsic limitations:
> We added a discussion of SWG’s limitations as follows. Since SWG jointly learns actions and weights, it lacks the flexibility of approaches with separate guidance networks that can be retrained independently. Joint learning also complicates optimisation, possibly requiring larger models. These issues can be mitigated by fixing a network submodule for weight learning.
>
> ### Suggested changes
>
> - We removed the equation number when these were not referenced in the text.
> - Although we agree with the Reviewer, we have decided to maintain the format “equation 6” —rather than Equation (6)---  as this referencing style is specified by the TMLR template when using eqref{eqn_label}. We trust the Reviewer will agree with us in the importance of abiding by the journal’s format.
> - We have incorporated additional information in Table I to support its relevance. Additionally, we now incorporate a diagram (Figure 1) to strengthen the message conveyed by Table I.
> - Thanks for this observation. The change in notation from Sec 2 to Sec 3 is intentional, since Sec 2 refers to DMs in the general case, while Sec 3 describes the application to the offline RL problem. Therefore, we emphasised the difference in these two settings via the notation.
> - Thanks, the sentence “The weighted behavior policy expression in equation 1 admits using DMs in offline RL” was changed to “The expression of the weighted behavior in equation 1 makes it possible to use DMs in the context of offline RL”
>
> References:
> [1] Hansen-Estruch et al. IDQL: Implicit Q-learning as an actor-critic method with diffusion policies. arXiv:2304.10573 (2023)
> [2] Kostrikov et al. Offline Reinforcement Learning with Implicit Q-Learning. ICLR (2022)
> [3] Ho et al. Denoising Diffusion Probabilistic Models. NeurIPS (2020)
> [4] Peng et al. Advantage-weighted Regression. arXiv:1910.00177 (2019)
> [5] Chi et al. Diffusion Policy: Visuomotor Policy Learning via Action Diffusion. IJRR (2024)

---

### Review · Reviewer_rDyL · 2025-10-27

**Summary Of Contributions:**

## Overview
This paper presents a novel exact guidance approach for diffusion policies called Self-Weighted Guidance (SWG).
Unlike previous methods that require a separate guidance function with often complex loss formulations, SWG eliminates the need for such explicit guidance by jointly learning actions and weights within a single diffusion model.
This enables the guidance score for sampling target policies to be computed directly from the model itself, resulting in a streamlined training and inference process.

## Experimental Validation
- SWG is validated on toy distributions, showing accurate sampling capabilities.
- The authors benchmark SWG and its variant SWG-R on standard D4RL datasets.
- Ablation studies are conducted to evaluate different weight formulations and model scalability.

## Strengths
- Delivers exact guidance directly from the model
- Simplifies the training procedure
- Broad applicability to weighted sampling tasks

## Weaknesses
- Relies on the choice of weight formulation
- Computing gradient through the diffusion model can be slow
- Requires retraining for changes in guidance
- Shows mixed performance on the D4RL benchmark tasks

**Audience:**

Yes

**Audience Explanation:**

The work addresses active research areas by introducing a novel self-weighted guidance mechanism for diffusion models in offline reinforcement learning, a topic relevant to both the generative modeling and RL communities. The paper demonstrates theoretical understanding through exact guidance derivations and provides rigorous empirical validation on standard benchmarks (D4RL, toy distributions, and ablation studies). These contributions are important for researchers interested in both offline RL and generative modeling using diffusion models. The paper aligns well with the interests and scope of TMLR’s readership.

**Broader Impact Concerns:**

Broader impacts of this work include the possibility of amplifying biases present in the underlying datasets, since policies learned using weighted behavior can perpetuate those. Additionally, training and deploying diffusion models with increased computation—as done in this paper—require substantial computational resources, leading to increased energy consumption and environmental impact.

**Claims And Evidence:**

Yes

**Claims Explanation:**

- The paper provides a clear derivation and theoretical proof for the SWG method, demonstrating the mechanism for exact guidance in diffusion models.

- It includes validation on toy distributions with known target energy functions, showing that SWG can sample accurately and consistently from these distributions.

- The method is tested on the D4RL benchmark suite, where results exhibit competitive performance in several challenging environments (Ant Maze, Franka Kitchen, Adroit Pen), although SWG shows only average results on locomotion tasks.

- The ablation studies on different weight formulations and model scaling further support the paper.

**Requested Changes:**

## Requested Changes

# Critical

- **Introduction:** Please moderate or clarify claims about SWG’s “top performance” in D4RL, as the results indicate it is not consistently superior across all datasets.

- **Section 2.2:** Rephrase the statement that diffusion models (DMs) are “highly expressive compared to classical neural architectures.” While DMs employ a different training paradigm, their underlying architectures are often similar to standard neural networks.

- **Algorithm 1:** Explicitly state which w function is used for dataset augmentation to improve clarity and reproducibility.

- **SWG-R Details:** Clarify whether the weights used in SWG-R for resampling are the diffused weights from the joint model or from athe separate weights model (w) .

- **Results:** Please define all dataset acronyms used in Table 2 (e.g., ME, M, MR, partial, mixed, complete, Avg) within the table caption or in a dedicated glossary/footnote for clarity. This will help readers unfamiliar with the terminology to interpret the results correctly.

- **Locomotion Results:** The paper downplays the importance of the saturated locomotion benchmarks, but the average or below–state-of-the-art performance of SWG-R on these tasks is notable.

# Strengthening contribution
 I consistently observe that SWG-R underperforms in the partial, mixed, mixed-replay, and diverse settings. Please discuss whether this underperformance could be attributed to the choice or quality of the pre-trained weight function, and elaborate on how dataset composition or weight function design might impact SWG-R’s effectiveness.

---

> ### Author Response · Authors · 2025-10-31
>
> Dear Reviewer,
>
> We truly appreciate your constructive review. We have updated the paper based on your suggestions and comments.
>
> ### Critical changes
>
> - **Introduction:** We updated the Introduction Section to use “competitive performance” instead of “top performance,” which makes our claims about SWG’s results on D4RL more moderate.
>
> - **Section 2.2:** We have rephrased “DMs are highly expressive compared to classical neural architectures” to “DMs are highly expressive compared to classical generative models.”
>
> - **Algorithm 1:** We now clarify in Algorithm 1 that our method uses the expectile weight model as the main weight model.
>
> - **SWG-R Details:** We added to the results that the weights used for resampling come from separate weight models.
>
> - **Acronyms for D4RL:** We added a list of  acronyms to the caption of Table 2.
>
> - **Locomotion results:** We agree with this comment and have  removed the part that downplayed the importance of the D4RL locomotion datasets.
>
> ### Strengthening contribution
>
> We appreciate this suggestion and have included a discussion of the sub optimal performance observed on mixed-action datasets (i.e., Mixed, Medium-Replay, and Diverse) in the conclusions . This performance is likely due to the chosen weight model and/or the architecture employed for the DM.

---

### Decision · Action_Editor_KLyx · 2025-12-05

**Recommendation:** Accept with minor revision

**Additional Comments:**

The manuscript was updated during the review process to include experimental results with an updated SWG+Resampling methodology. Some additional work in the manuscript is necessary to fully integrate these details -- e.g. the end of Section 4 states such resampling is not proposed / used in the work. I believe these modifications will be relatively minor, so advocate for acceptance with minor revision.

**Audience:**

Yes

**Audience Explanation:**

All reviewer agree there is an audience within the offline reinforcement learning community that would be interested in the proposed work and the AE agrees. The proposed methodology is well-positioned among related methods and provides some simplifications in the training process.

**Claims And Evidence:**

Yes

**Claims Explanation:**

Reviewers are generally supportive about the claims of the submission after the revision due to improved results and some changes to scope of claims. The updated SWG-Resampling is more competitive with prior art while maintaining the core ideas of the manuscript.